# Fake News and Covid-19 in Italy: Results of a Quantitative Observational Study

**DOI:** 10.3390/ijerph17165850

**Published:** 2020-08-12

**Authors:** Andrea Moscadelli, Giuseppe Albora, Massimiliano Alberto Biamonte, Duccio Giorgetti, Michele Innocenzio, Sonia Paoli, Chiara Lorini, Paolo Bonanni, Guglielmo Bonaccorsi

**Affiliations:** Department of Health Science, University of Florence, 50134 Florence, Italy; giuseppe.albora@unifi.it (G.A.); massimilianoalberto.biamonte@unifi.it (M.A.B.); duccio.giorgetti@unifi.it (D.G.); michele.innocenzio1@stud.unifi.it (M.I.); sonia.paoli@unifi.it (S.P.); chiara.lorini@unifi.it (C.L.); paolo.bonanni@unifi.it (P.B.); guglielmo.bonaccorsi@unifi.it (G.B.)

**Keywords:** social media, health literacy, Covid-19, coronavirus, fake news, BuzzSumo

## Abstract

During the Covid-19 pandemic, risk communication has often been ineffective, and from this perspective “fake news” has found fertile ground, both as a cause and a consequence of it. The aim of this study is to measure how much “fake news” and corresponding verified news have circulated in Italy in the period between 31 December 2019 and 30 April 2020, and to estimate the quality of informal and formal communication. We used the BuzzSumo application to gather the most shared links on the Internet related to the pandemic in Italy, using keywords chosen according to the most frequent “fake news” during that period. For each research we noted the numbers of “fake news” articles and science-based news articles, as well as the number of engagements. We reviewed 2102 articles. Links that contained fake news were shared 2,352,585 times, accounting for 23.1% of the total shares of all the articles reviewed. Our study throws light on the “fake news” phenomenon in the SARS-CoV-2 pandemic. A quantitative assessment is fundamental in order to understand the impact of false information and to define political and technical interventions in health communication. Starting from this evaluation, health literacy should be improved by means of specific interventions in order to improve informal and formal communication.

## 1. Introduction

On 11 March 2020, only ten years after the H1N1 swine influenza pandemic [1], a new respiratory virus surged globally to gain recognition as a pandemic disease [2]. Initially, the disease was mistakenly considered not substantially more disruptive than influenza [3]. This is a respiratory RNA virus from the family of Coronaviruses. The “Coronavirus Study Group” from the International Committee on Taxonomy of Viruses called it SARS-CoV-2 (Severe acute respiratory syndrome coronavirus 2), due to the similarities with the SARS virus (SARS-CoV) [4].

The first cases in Italy were reported on 30 January 2020, when a couple of tourists tested positive in Rome [5]. Since the beginning of March, data regarding the Covid-19 (Coronavirus Disease-2019) pandemic began to dominate newscasting and news outlets in Italy. Simultaneously, the scientific community (biologists, epidemiologists, and clinicians) was buzzing fervently and generating a tremendous amount of articles as they began to learn more about the virus.

By mid-July 2020, the pandemic accounted for 13,486,823 cases, with 581,965 deaths and 7,876,115 recovered patients worldwide [6]. The socioeconomic consequences of the pandemic have been devastating and countless, and for the most part they are to be evaluated yet. Such an overwhelmingly devastating infectious disease was not foreseen in the popular imagination. SARS-CoV-2 managed to rapidly shift a paradigm that had been firmly established for years (at least in developed countries)—non-communicable diseases are not the only concern for the global community, as we used to think before. It has been a complete upheaval of the norm. Decision makers (governments and institutions) were left in a difficult and dangerous situation in terms of risk management and risk communication regarding infection rates and restrictive measures of containment that were imposed to slow down the spread of the virus [7]. Top-down communication was at times ineffective and inefficient in a multitude of situations, resulting in social unrest, hoarding of groceries, and chaotic exoduses from cities in quarantined areas [8,9]. Within this context, it is also necessary to refer to the readability of the information about Covid-19. In fact, a cross-sectional study conducted by Szmuda et al. through the use of validated readability tests applied on articles found on the Internet showed that online educational articles on Covid-19 provide information that is too difficult to be understood by the general population [10]. This fatally helps (or even boosts) the spread of false information.

In such times, people can be prone to scapegoating; religion, ethnics, wealth, and gender can all become social dividers that can heighten the social unrest and uneasiness [11]. The uneasiness and restlessness of the population have been enhanced by the restrictive and preventive measures, which can become a stressor in daily activities.

Conspiracy theories and alternative facts find fertile ground on the Internet, feeding off the restlessness and helplessness of people, frustrated firstly because they could not foresee the disaster and secondly because they are unable to indicate the culprit of such tragedy. The void left by the absence of certainty can make any kind of explanation plausible and reasonable to the eyes of the general population. 

Although there is still not a clear definition of this false information, or so-called “fake news”, we can affirm that such articles are born from a piece of news that is deliberately built from false and mendacious information but looks plausible and factual to the eyes of the reader. The aim is to manipulate and mold public opinion. While a factual news story is created by a journalist, fake news is a product of both the writer and the audience, inasmuch as it only exists if it is perceived to be real [12]. The Internet and social media are remarkably fertile grounds for alternative facts and fake news [13], since there is not a sufficiently coherent fact checking system on the Internet [14]. Real and fake news blend together and it is up to the user to filter through a massive mound of news stories according to one’s own capabilities, often influenced by: (1) cognitive biases (confirmation bias, cherry picking), (2) the lack of willingness to fact check, (3) rarely adequate digital literacy, and (4) incohesive or entirely absent health literacy (i.e., the ability to obtain, read, understand, and use healthcare information in order to make appropriate health decisions and follow instructions for treatment) [15,16]. UNESCO (United Nations Educational, Scientific and Cultural Organisation) recently publicized a handbook about the issues affecting journalism and information. One of the purposes is warning that the term “fake news” does not have a straightforward or commonly understood meaning. In fact, “news” means verifiable information in the public interest; information that does not meet these standards does not deserve the label of “news”. Therefore, “fake news” is an oxymoron that lends itself to undermining the credibility of the “real” news. Thus, it would be better to use other terms to describe a phenomenon where there is a spreading of untruths via social media and the Internet, involving disinformation (false information deliberately created to harm a person, social group, organization, or country), misinformation (false but not created with the intention of causing harm), and mal-information (based on reality, used to inflict harm on a person, social group, organization, or country) [17].

A recent study claims that 82% of Italians are not able to recognize fake information [18]. The Internet is still an obscure tool for most of the Italian population. In 2016, only 37% of people aged 16–34 and 23% of people aged 45–54 years old claimed they possessed adequate skills concerning the Internet [19]. Another pilot study in 2019 showed that only 62.3% of the studied population had adequate skills concerning reasoning and autonomy, while the rest had inadequate (11.2%) or problematic skills (26.5%) when it came to health literacy [20].

Digital and health illiteracy can account for most of the inaccurate news stories depicting the cause of the pandemic, whether suggesting it is a virus engineered purposely in a Wuhan laboratory [21], a biological military weapon [22], or a result of 5G (5^th^ Generation) technology being implemented in many countries [23]. Some people were singled out as culprits guilty of causing the pandemic, such as Bill Gates [24]. We can even find news stories claiming that as-yet undiscovered vaccines for SARS-CoV-2 are harmful and useless [25,26]. 

Such fake stories are intentionally built to rapidly and tumultuously spread over the Internet and social media. Their popularity is rapidly increasing, such that retractions are always published too late. As many philosophers used to say, “a lie can run around the world before the truth can get its boots on” [27].

The seriousness of the situation was duly described in a recent issue of The Lancet, which published a statement in support of scientists, public health, and medical professionals, condemning all theories validating a non-natural origin of the SARS-CoV-2 virus and citing a list of articles proving the enormous amount of studying that was going into the sequencing of the genome of this novel pathogen [28,29,30,31,32].

The Lancet again published an article in 2015 denouncing the still underestimated inadequateness of health literacy in Europe [16], and more recently it published an article by Paakkar that underlines how health literacy is a serious concern in light of the recent events surrounding the Covid-19 pandemic [33]. This phenomenon is a challenge not only for older adults, but also for young people and university students. A recent study demonstrates that less than 50% of students have high levels of health literacy, and that they learnt about the Covid-19 pandemic via social media and the Internet [34]. In order to understand the reason for the success of fake news in such situations, it is necessary to analyze the impact of the quarantine on the Italian population and the psychological strain that it caused. An online survey conducted in Italy during the lockdown established the prevalence of psychiatric symptoms linked to the pandemic in the general population [35], such as anxiety and depression. This is closely related to the spread of fake news, because such articles can be more successful when the population is experiencing a stressful psychological situation. Moreover, during the lockdown the population spent more time on the Internet and social media, so the impact of false information was higher than normal. In human history, such an abundance of health information from more or less trustworthy sources has never been seen before, and without an appropriate cultural background and literacy, it is difficult for the public to understand that the best scientific knowledge on Covid-19 needs time to grow and that there is a need to critically assess available information [36].

“Fake news” has inevitably and indubitably influenced health communication in the Covid-19 emergency and it is clear that it might continue to do so for the foreseeable future, contributing to social unrest and uneasiness. The aim of this study is to measure—using nine specific key words—how much false and true information have circulated in Italy in the period between 31 December 2019 and 30 April 2020, as part of the phenomenon called infodemia [37]. More generally, we hope to shed some more light on the fake news phenomenon and the reasons for its popularity.

## 2. Materials and Methods

We used the BuzzSumo pplication [38] in order to gather the most shared links or posts on the Internet and social media related to SARS-CoV-2 and the Covid-19 pandemic. The BuzzSumo Application is one of the most popular social media trend analysis tools, which is often used in marketing research in order to highlight content that has a very strong engagement score (i.e., number of shares, links, comments, and backlinks), either on the Internet or social media. 

BuzzSumo gathers data across the social media platforms Facebook, Pinterest, Reddit, and Twitter to generate a list of article links with the highest online engagement. Engagement is defined as the total number of interactions that users have with a particular article link, including actions such as “liking”, “commenting”, and “sharing” on social media [39,40]. Since BuzzSumo is free of charge when using the 7-day trial feature, we decided to use this tool to make the study more transparent and reproducible.

We restricted our search to three specific date ranges: (1) 31 December 2019–19 February 2020; (2) 20 February 2020–10 March 2020; (3) 11 March 2020–30 April 2020. The data ranges were selected according to self-evident pivotal moments in the events timeline of the Covid-19 pandemic in Italy (details in Table 1) [41,42,43].

The BuzzSumo Application allows web content research through the use of keywords. We selected 9 different search bars, each of which was composed of two keywords.

The search bars we applied always contained the word “coronavirus” and one other specific associated term, which was different for each search bar. The terms we inserted after “coronavirus” were: “vaccine” (in Italian “vaccino”), “origin” (“origine”), “laboratory” (“laboratorio”), “plot” (“complotto”), “HIV” (“HIV”), “vitamin C” (“vitamina C”), “vitamin D” (“vitamina D”), “garlic” (“aglio”), “5G” (“5G”). The keywords were chosen according to the most frequent topics of the “fake news” that spread during the Covid-19 pandemic. Such a specific set of keywords was purposefully chosen because it reflected our intent to only look for health-related news stories about Covid-19. The 9 keywords were chosen in a consensus meeting of the research group, since they were the most likely to uncover health-related false information using the BuzzSumo search engine, and specifically fake news that would not meet our exclusion criteria.

Searches were filtered by time and language (Italian), and for each one of the 9 terms a search was conducted on the three specific date ranges we exposed previously. Therefore, we conducted 27 separated searches.

We noted the total findings that the BuzzSumo application was able to provide. We manually revised each one of the top shared results (links, posts, videos, articles), looking for false information (disinformation, misinformation, and mal-information) and incomplete or false coverage of Covid-19-related content, as well as science-based news and articles about those topics. We applied strict exclusion criteria when conducting our review. We excluded content that was merely a report of events, focusing on medical and scientific subjects. An article was immediately excluded when the content did not deal specifically with health or science, i.e., the focus may have been on the socioeconomic consequences of the pandemic, which was a topic we excluded from our fake news review. Moreover, we decided to include articles that were reporting fake news about SARS-CoV-2 to give a sense of reasonable doubt, i.e., claiming that there were certain allegations about a certain topic but pointing out that such allegations had no evidence whatsoever supporting them. Articles were classified as fake news if the content was not supported by scientific literature or when the data reported were used to make inappropriate conclusions about SARS-CoV-2. The contents of the most frequent fakes found for each topic are reported in Table 2. We concluded each search when all of the findings were examined or once we had reviewed 100 articles by adding the number of false articles to the number of science-based articles. Each link was independently reviewed by two of the authors. In case of disagreement between the authors, a third author was involved in the review. For each search, we noted the number of excluded articles, the number of fakes, and the number of science-based articles examined. Moreover, we noted the amount of engagement with the fake and science-based articles that we considered for our study. We also noted the source of every “fake” article reviewed. Data were collected and additionally analyzed with Microsoft Excel.

## 3. Results

Overall, we reviewed 2102 articles that were generated by our keywords search on BuzzSumo. The analyzed topics attracted public attention with unequal distribution:—laboratory, vaccine, and 5G articles accumulated more shares than other topics.

Altogether throughout the three periods, links that contained untrue information were shared 2,352,585 times, accounting for about 23.1% of the total shares of all the articles reviewed. The topics most contaminated with fakes were vitamin D (89.4%), HIV (77.8%), and garlic (71.2%). A synoptic comprehensive analysis of the data we gathered from the keyword search is shown in D. 

Comprehensive and synoptic data analysis of all content analyzed are reported in Figure 1. Moreover, we summarized the distributions of the numbers of shares and new articles in Table 3.

The increasing percentage of “fake news” for numerous topics of research seems directly linked to specific events, for example when popular or well-established sources start supporting theories about SARS-CoV-2 without presenting evidence for their claims.

As a significant example, we can analyze the fluctuations of this false information when searching “coronavirus HIV”. In the first period, there was 1 fake article with 767 total shares, in the second period there was 1 fake article with 3 total shares, and finally in the third period there were 15 fake articles with 72,715 total shares. Such a rapid increase can be traced back to some statements about the origin of the virus by the 2008 Nobel Prize for Medicine winner Dr. Montagnier, which were not supported by any evidence. 

Something similar seemed to happen when searching for “coronavirus laboratory” (“coronavirus laboratorio”). In the first period, this term accounted for 55 fake articles and 1,039,224 total shares, while in the second period only 23 fake articles and 12,972 total shares were counted. The first period coincided with claims by a news reporter from a popular newscast about the alleged laboratory origin of the virus in Wuhan. 

The trend even more evident when searching “coronavirus garlic” (“coronavirus aglio”), where in the first and second periods 0 fake articles and 0 shares emerged, while in the third period 7 fake articles received 5284 total shares, whereby garlic was described as a miraculous treatment for Covid-19 without presenting any evidence whatsoever [44]. 

When searching for “coronavirus vaccine” (“coronavirus vaccine”), the total numbers of shares between the first and third period increased from 14,106 to 206,900, an 18.5-fold increase, following the more extensive media coverage about vaccine research for SARS-CoV-2. 

Similar patterns of fluctuations of false information were found for the other keywords for topics dealing with prevention or treatment of Covid-19. In conclusion, it was concluded with no exception for any given keyword that fake news in the Covid-19 pandemic succeeded multiple times in overshadowing formal verified news. This phenomenon was in many cases not only limited to the gross number of new articles produced, but also the number of shares on social media. 

Fake news indubitably and perceivably affected health communication during the Covid-19 pandemic. What this study tried to achieve was to quantitively gauge the amount of fake news and to decipher the common patterns and mechanisms that underlie the tumultuous spread of fake news on social media. 

The data gathered was also able to show how apparently few fake news story (i.e., a low number of new articles out of the total percentage) can account for a vast majority of shared news stories on social media (i.e., high number of shares as a percentage). This is very self-evident when looking at the data for the keyword “garlic” (“aglio”), where 18% of the news stories that were classified as fake accounted for almost 70% of the shares on social media. 

## 4. Discussion

The spread of false information, or so-called “fake news”, in non-official communication can ultimately be considered a very disruptive and dangerous phenomenon that can profoundly undermine health and risk communication, particularly in an emergency, such as the one we are living through.

The number of news stories regarding the Covid-19 pandemic continued to increase through the three periods we considered, which is easily understandable, as the virus started to spread dramatically throughout the globe and newscasts increasingly focused on that subject. As the number of news stories increased, so did the number and percentage of untrue information about the Covid-19 pandemic. 

Our observations regarding the percentage distribution between shares and news were particularly interesting (Table 3). In 6 out of 9 search topics (“vitamina C”, “vitamina D”, “aglio”, “5G”, “laboratorio”, “HIV”), the percentage of shares for fake articles was greater than that of verified new articles. For example, regarding the keyword “HIV”, the percentage of fake regarding was 11.1% (therefore, the percentage of real news was 88.9%). This ratio is not mirrored when observing the percentages of shares for both fake and real news. Shares of “fake news” accounted for 77.8%, while shares of real news accounted for 22.2%. This means that on average, “fake news”—standardized for the same number of total news results—seems to have a higher number of shares when compared to real news; that is, a much higher likelihood of being shared and known. 

The tendency to produce false content is born and spread for a variety of reasons. One of the main reasons is when such news stories are conveyed by authoritative figures, as in the case of the lab incident in Wuhan while testing an HIV vaccine [21]. When this scenario finds the support of well-known figures, it becomes popular and quasi-real, even without being supported by data or evidence, feeding off the idea that there is some sort of plot to silence people that are perceived as menacing or challenging the status quo in the scientific community. 

Other reasons can be attributed to the fact that Covid-19 is not only a health issue, but also a large-scale socioeconomic disaster, which is so deeply impactful that it has monopolized international newscasts. The pandemic contributed further complicated the political relations between the US and China, the two biggest economies in the world, to the point that some are calling it a new cold war [45]. In this context, fake news has become a political tool used to discredit either country, inflaming already difficult diplomatic relations [46,47].

The last and perhaps more important reason can be found in the scapegoating phenomenon. One dramatic aspect of the response to the pandemic was the desire to assign responsibility by identifying the culprit, which is a recurrent situation in history. For instance, during the Black Death in 1347, this role was attributed to the Jewish people—they were accused of poisoning the water wells with the intent of killing all Christians. The same happened in the 1630 plague—in “The Betrothed”, Manzoni described how the blame was casted upon disease spreaders, astral influences, or poisonous exhalations, the so-called miasmas [48].

The scapegoating phenomenon exploits existing social divisions of religion, race, ethnicity, class, or gender identity, fueling social conflict and friction between governments and the general population [49]. 

To understand the reasons for the popularity of “fake news”, it will be necessary to deeply analyze and better understand the psychological mechanisms underlying its success.

The universe of “fake news” is heterogeneous, but recurring elements are identifiable in terms of the context, methods of diffusion, and the target population that the fake news is addressed to. 

As for the context, social networks have proven to be the most fertile ground for the spread of false facts. They offer a great space for profiles and pages that are created with the precise intent of generating and spreading fake news [50,51]. 

False facts are usually presented with some distinguishing features—the content is mostly represented in the title, while the page that shows the entire text (once the title is clicked) often shows only a bare repetition of the key concept [52].

Titles are usually short with “clickbait” features, written in larger fonts, with a sly use of colors and photos. The aim is to appeal and attract an audience that is used to ratiocinating intuitively, through “heuristics” (mental shortcuts that simplify the analysis of the text) that lead to “biases” (cognitive distortions that lead to misinterpretation of the text) [53].

The intrinsic features of the target population also play a role in the spread of “fake news”. Recent studies show that subjects prone to depression, disappointment, suspicion, and religious fundamentalism are more susceptible to this type of content [54]. Age is also positively correlated with the probability of believing in and sharing false facts. 

Although the key psychological mechanisms still require thorough research, many experiments point to a correlation with the scores obtained in tests that assess analytical skills, such as the cognitive reflection test (CRT) [55]. CRT is a simple test that attributes a low score to a subject that shows “intuitive” reasoning (that is more impulsive and emotional) and a high score to a subject that shows predominantly “analytical” reasoning (that gives more thoughtful responses), according to the dual-process theory [56]. Subjects who are more inclined to believe in “fake news” have lower scores, showing a predominantly “intuitive” style and being guided by the sensations aroused by the news itself, by biases and heuristics, and by interrupting fact-checking processes [57]. Intuitive reasoning can especially be found (but not always) in older people or in those with lower educational attainment, while the “analytical” style tends to be applied more frequently in people with higher educational level and younger age [58].

Social media also allows for massive repetition of “fake news” content, creating some sort of “echo chamber” phenomenon. The constant repetition of the same false information can trigger a feeling of familiarity in the reader, determining a further drop in attention and fact-checking. This all finally results in a greater acceptance of the content propagated by the “fake news” story—the more false content is shared, the more a user is led to believe it [58,59,60,61,62].

In this scenario, several research studies are aimed at finding strategies to reduce the impact of “fake news” and inaccurate information. This goal could be achieved by limiting the repetition and massive sharing of inaccurate content. However, it is a difficult approach, since repetition and echo-chamber effects are intrinsic characteristics of social media. 

Another possible solution would be to immediately deny and discredit “fake news”. This strategy could work provided that the retraction is rapid and effective. It has been proven that even after pointing out incorrect information, the initial influence of false information cannot be undone. According to a study that investigated the persistence of feelings, some individuals, especially those with lower cognitive abilities, may continue to foster negative feelings or express discomfort towards a certain topic even after disconfirmation of the false information [63].

A strategy that has been more successful in the fight against “fake news” promotes the use of nudging; it would be enough, in fact, to remind the subjects to pay attention in assessing the veracity of some statements before reading them in order to improve their ability to understand which news is false and which is not [64]. Furthermore, a greater comprehension, acknowledgement, and application of health literacy can support policy action on multiple levels to address major public health challenges. Health literacy should be built deliberately as a population-level resource and community asset, and the relevance of mass and social media suggests including them in planning communication interventions related to environmental health and in verifying their results [65,66]. The latter studies suggests that the route to defeat “fake news” may hopefully be less complicated than we think.

## 5. Conclusions

Our study illustrates the “fake news” phenomenon in the Covid-19 pandemic. The spread of false content related to health communication was a very prominent feature throughout the three periods we considered, which had a deep and meaningful impact upon the general population. It was an element that undoubtedly undermined informal communication during the health emergency. 

Health-related fake articles have been thoroughly studied and analyzed, especially in the last few years, when they became a disruptive element in the conversation about vaccines. It was not a surprise to encounter false facts during the Covid-19 pandemic.

Such mendacious and false new articles are easily traced back to their source and probably do not require more top-down, specific qualitative studies about their origin and nature. However, we believe that a quantitative and qualitative evaluation is fundamental in order to assess the real impact of fake news and to start defining political and technical interventions in health communication.

In conclusion, we believe that there are two paths to minimize the impact of “fake news”. On the one hand, we must improve health and digital literacy—it has been demonstrated that a low level of health literacy brings people who have suspected symptoms related to Covid-19 to feel more stressed and depressed than people who have higher levels of health literacy [67], which also happens to medical students in terms of their fear for Covid-19 [68]. Having better literacy is helpful in fighting the fear and stress related to the pandemic. On the other hand, we must start favoring better informal communication and more organized formal communication. Along these lines, it is hoped that social media companies—arguably some of the most important communication platforms today—will be able to further improve, strengthen, and reinforce their policies against “fake news”.

## Figures and Tables

**Figure 1 ijerph-17-05850-f001:**
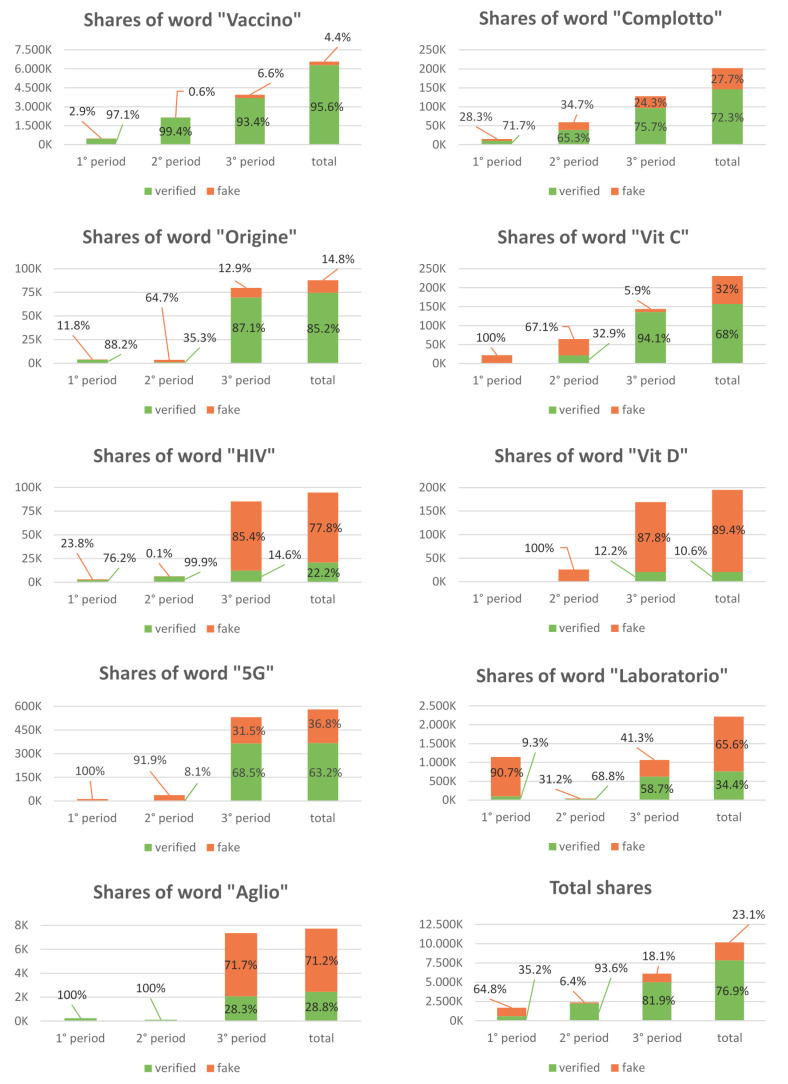
Comprehensive and synoptic data analysis of all content analyzed (vaccine = vaccino; origin = origine; laboratory = laboratorio; plot = complotto; HIV = HIV; vitamin C = vitamina C; vitamin D = vitamina D; garlic = aglio; 5G = 5G).

**Table 1 ijerph-17-05850-t001:** Main events of Covid-19 pandemic in Italy.

Periods	Events
Period 1: 31 December 2019–19 February 2020	Wuhan Municipal Health Commission, China, reported a cluster of cases of pneumonia in Wuhan, Hubei Province. A novel coronavirus was eventually identified.
Period 2: 20 February 2020–10 March 2020	“Patient 1” identified in Codogno, Italy
Period 3: 11 March 2020–30 April 2020	Lockdown is extended to all Italy

**Table 2 ijerph-17-05850-t002:** Contents of the most frequent fake news.

Key Words Italian (English)	“Fake News”
Origine (origin)	The outbreak was a result of an accidental release from a Wuhan laboratory Sars-Cov-2 does not have an animal origin
Laboratorio (laboratory)	Sars-Cov-2 was engineered in a Wuhan laboratory
Complotto (plot)	Sars-CoV-2 was created by governments Sars-Cov-2 is a biological weapon
HIV (HIV)	Sars-Cov-2 was engineered in a laboratory when researching a vaccine for HIV
Vitamina C (vitamin C)	Vitamin C supplements protect against the Sars-Cov-2 infection
Vitamina D (vitamin d)	Vitamin D is an effective therapy against Covid-19 A high level of vitamin D in the blood is effective in preventing Covid-19
Aglio (garlic)	People who eat large quantities of garlic are not infected by Sars-Cov-2
5G (5G)	Correlation between 5G technology and the spread of Sars-CoV-2

**Table 3 ijerph-17-05850-t003:** Percentage distributions of total links and shares for each search. Period: 31 December 2019–30 April 2020.

Key Words Italian (English)	No. News (%)		No. Shares (%)	
	Fake	Verified	Fake	Verified
Vitamina C (vitamin C)	24.0%	76.0%	32.0%	68.0%
Vitamina D (vitamin D)	79.6%	20.4%	89.4%	10.6%
Aglio (garlic)	18.0%	82.0%	68.4%	31.6%
5G (5G)	32.8%	67.2%	36.8%	63.2%
Vaccino (vaccine)	7.7%	92.7%	6.6%	93.4%
Complotto (plot)	29.3%	70.7%	27.7%	72.3%
Origine (origin)	37.3%	62.7%	14.8%	85.2%
Laboratorio (laboratory)	52.8%	47.2%	65.6%	34.4%
HIV (HIV)	11.1%	88.9%	77.8%	22.2%
Total	31.9%	68.1%	23.1%	76.9%

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
