# Peer review of "Fake News and Covid-19 in Italy: Results of a Quantitative Observational Study"

_ijerph, 2020, doi:10.3390/ijerph17165850_

Round 1

Reviewer 1 Report

The proposed paper deals with a very important topic and it is based on recent data. I would suggest some improvements and rewritings so the presented information is clear and the impact of the paper is higher.

  • Table 1: It is not clear what are the milestones. You mention one specific moment for each of the period, however, it is not clear if the moment represents the beginning or end of the period.
  • You mention the absence of scientific data in the article as a criterium to code a link/article as fake news. This seems to be unfortunate to me – you cannot say it’s fake news just because there are no scientific studies mentioned in the article.

  • Each link was reviewed by two of the authors; what if there was no agreement in coding? Was there any intercoder reliability test prior to the coding which could ensure that the agreement of two authors is not just coincidence? How did you solve the situation authors coded a link differently from each other? It is not clear.

  • There are statements in the text which are not accompanied by a reference. For example, in the Results section, page 5, you mention Dr Montagnier’s statement, however, this is not referenced so readers have no chance to read it or verify it.

  • The results are not sufficiently explained. They are presented as portions of shares, with almost no context. What did you find? What are the results implying? What are the lessons learned or points to takeaway?

  • Some of the thoughts presented in the Discussion section seems to be too obvious – it is logical there is more fake news on the linkage between coronavirus and HIV than real news since there is no relationship proven. It is a question if the points addressed in the Discussion are not too far from the focus of the paper. Of course, they are related to the fake news in a broader sense, however, they are not directly connected to the aim of the research.

  • Some parts of the paper's internal structure could be also slightly improved, they read fragmented.

Author Response

Point 1

  • Table 1: It is not clear what are the milestones. You mention one specific moment for each of the period, however, it is not clear if the moment represents the beginning or end of the period.

Response 1

Dear Revisor, in the timeline of the novel "coronavirus in Italy" we chose three events that, in our opinion, had a remarkable importance for the population. The three events are: 1) official communication from China about the identification of the new virus Sars-Cov-2 (31/12/2019); 2) patient 1 in Italy is identified in Codogno (20/2/2019); 3) Lockdown is extended to all Italy (11/3/2020). Please, consider that we decided to begin our research on the first day of May. Therefore, these four dates (3 events + the last day before the beginning of our research) divide the timeline into three specific date ranges which are those one reported in Table 1. Therefore, the three events reported on Table 1 represent the beginning of each date range.

Point 2

You mention the absence of scientific data in the article as a criterium to code a link/article as fake news. This seems to be unfortunate to me – you cannot say it’s fake news just because there are no scientific studies mentioned in the article. Each link was reviewed by two of the authors; what if there was no agreement in coding? Was there any intercoder reliability test prior to the coding which could ensure that the agreement of two authors is not just coincidence? How did you solve the situation authors coded a link differently from each other? It is not clear.

Response 2

Thank you very much for the comment. Each link review was performed by two authors, but in case of disagreement between these two, another author was included in the review. We added a line in “materials and methods” section to better explain this passage.

Regarding the line that reports “article was classified as false if was not supported by scientific literature or when the data reported were used to make inappropriate conclusions about the SARS-CoV-2”, we classified as “false” a fact that was verified by a scientific study as statistically NOT associated to Sars-Cov 2, or that clearly made captious interpretation of the results of a scientific study not completely related to Sars-Cov 2. In most cases it was really clear which fact was fake and which not and with the “double review” we classified all the links in the database.

Point 3

There are statements in the text which are not accompanied by a reference. For example, in the Results section, page 5, you mention Dr Montagnier’s statement, however, this is not referenced so readers have no chance to read it or verify it.

Response 3

Thank you for your suggestion, it is true: we reported the reference of Dr Montagnier's statement in the Introduction section (page 2, reference n° 20) but we did not report it again in the Results section so we modified the manuscript and we added it.

Point 4

The results are not sufficiently explained. They are presented as portions of shares, with almost no context. What did you find? What are the results implying? What are the lessons learned or points to takeaway?

Response 4

Thank you for your feedback. In order to answer to your revision, we have updated with hopefully more in-depth and clear information the “results” portion of the article. What we tried to clarify can be summarized in two points:

  • show how fake news overshadowed and towered above verified news in social media and how such results were achieved not always because of quantitative superiority in the gross number of news stories but also because of undeniable popularity
  • show how such popularity usually relies on recurrent themes and patterns both in the way how fake news stories are perpetrated throughout social media and how they are produced and then construed.

Reviewer 2 Report

This research is interesting and is well-founded. However, there are two things that I think they are to detail in the document: 

1) When did the authors speak about articles: To which it refers? Newspaper? Post on Social Media? Fake newspaper? There are different supports of information and disinformation.  

2) UNESCO defines the concepts related to disinformation and specifies to avoid "fake news". "It avoids assuming that the term ‘fake news’ has a straightforward or commonly- understood meaning. This is because ‘news’ means verifiable information in the public interest, and information that does not meet these standards does not deserve the label of news. In this sense then, ‘fake news’ is an oxymoron which lends itself to undermining the credibility of information which does indeed meet the threshold of verifiability and public interest – i.e. real news. ". This definition I think to affect your work. https://en.unesco.org/fightfakenews

Author Response

Point 1

When did the authors speak about articles: To which it refers? Newspaper? Post on Social Media? Fake newspaper? There are different supports of information and disinformation.

Response 1

Dear reviser, thank you so much for your opinion and questions. In order to clarify your doubts about the news sources of this paper we decided to give you a deeper explanation of the search we applied.

The research on BuzzSumo Application was conducted using the selection criteria of the sources of the application itself. We decided to apply no search criteria in order to maintain this study more reproducible. According to this intention, the whole sources identified by the application were accepted.

Among the sources of news identified in our search are included newspaper articles and fake newspaper articles, as well as YouTube videos. The BuzzSumo application instead does not considered the personal posts that people could write on social media (with the only exception of YouTube Videos shared on them).

The research was conducted in order to gather the links with higher online engagement, defining it as the whole number of interactions made by users on the analysed link ( among them are considered actions as “like”, “comment”, “share”).

Point 2

UNESCO defines the concepts related to disinformation and specifies to avoid "fake news". "It avoids assuming that the term ‘fake news’ has a straightforward or commonly- understood meaning. This is because ‘news’ means verifiable information in the public interest, and information that does not meet these standards does not deserve the label of news. In this sense then, ‘fake news’ is an oxymoron which lends itself to undermining the credibility of information which does indeed meet the threshold of verifiability and public interest – i.e. real news. ". This definition I think to affect your work. https://en.unesco.org/fightfakenews

Response 2

Thank you for your observation, we really appreciated it. We added Unesco's reference and a citation from the UNESCO Handbook, we think that it is useful, especially in the Introduction section, where we talk about the meaning and origin of “fake news”. The purpose of the study is, in fact, to analyze the impact of all false facts and informations circulating on the social media; we included “false information”, “inaccurate content” and other similar terms in the article to better merge the UNESCO point of view.

Reviewer 3 Report

Fake news born from news that is deliberately built from false information, but seems plausible and factual to the reader, in order to manipulate public opinion. Fake news has inevitably and indubitably influenced health communication also in the Covid-19 emergency.

In this context, the aim of the manuscript under review is to measure how much fake news and true news have circulated in Italy in the period between December 2019 and April 2020. Currently, few evidences concerning fake news and Covid-19 are available in Italy.

The study methodology is appropriate and the paper appears very interesting. The ethic issue is not applicable.

In my opinion what is lacking is only a short context analysis related to COVID-19 Knowledge in subgroups of Italian people (eg. Gallè F, et al. Understanding Knowledge and Behaviours Related to CoViD-19 Epidemic in Italian Undergraduate Students: The EPICO Study. Int J Environ Res Public Health. 2020;17(10):E3481) and on what are the effects of emergency and fake news on the same population (eg.: Mazza C et al. A Nationwide Survey of Psychological Distress among Italian People during the COVID-19 Pandemic: Immediate Psychological Responses and Associated Factors. Int J Environ Res Public Health. 2020;17(9):3165). Introduction can be enriched accordingly.

Author Response

Dear revisor, thank you for your suggestion. We added in the introduction a short analysis of the effect of the lockdown on the Italian population mentioning one of the articles you suggested.